# Spatial changes in soil stable isotopic composition in response to carrion decomposition

Sarah W. Keenan[1,2], Sean M. Schaeffer[1], and Jennifer M. DeBruyn[1]

[1] University of Tennessee, Department of Biosystems Engineering and Soil Science, 2506 E.J.

Chapman Drive, Knoxville, TN 37996

[2] Current address: South Dakota School of Mines and Technology, Department of Geology and

Geological Engineering, 501 E. St. Joseph Street, Rapid City, SD 57701

Corresponding authors: sarah.keenan@sdsmt.edu, jdebruyn@utk.edu



**Abstract**

Decomposition provides a critical mechanism for returning nutrients to the surrounding environment. In terrestrial systems, animal carcass, or carrion, decomposition results in a cascade of biogeochemical changes. Soil microbial communities are stimulated, resulting in

transformations of carbon (C) and nitrogen (N) sourced from the decaying carrion soft tissues, changes to soil pH and electrical conductivity as microbial communities release $CO_2$ and mineralize organic N, and significant changes to oxygen availability. Over time, microbial communities transform ammonium to nitrate and potentially $N_2O$ through nitrification and denitrification. While many of the rapid changes to soil biogeochemistry observed during carrion

decomposition return to background or starting conditions shortly after soft tissues are degraded, some biogeochemical parameters, particularly bulk soil stable $\delta^{15}N$ isotopic composition, have the potential to exhibit prolonged perturbations, extending for several years. The goal of this study was to evaluate the lateral and vertical changes to soil stable isotopic composition one year after carrion decomposition in a forest ecosystem. Lateral transects extending 140 cm from three

decomposition "hotspots" were sampled at 20 cm intervals, and subsurface cores were collected beneath each hotspot to a depth of 50 cm. Bulk soil stable isotopic composition ($\delta^{15}N$ and $\delta^{13}C$) indicated that one year after complete soft tissue removal and decay, soils were significantly $^{15}N$-enriched compared to control soils up to 60 cm from the hotspot center, and enrichment extended to a depth of 10 cm. Our results demonstrate that carrion decomposition has the potential to

result in long-term changes to soil biogeochemistry, up to at least one year after soft tissue degradation, and to contribute to bulk soil stable isotopic composition.





## 1 Introduction

Nutrient hotspots are introductions of carbon (C) and nitrogen (N)-rich compounds into

an ecosystem, resulting in elevated reaction rates compared to surrounding regions (McClain et

al., 2003). For terrestrial and aquatic systems, hotspots may be sourced from fallen trees (Lodge

et al., 2016), annual deposition of deciduous leaves (Vidon et al., 2010), animal scat (Erskine et

al., 1998; van der Waal et al., 2011), or animal carcasses (Parmenter and Lamarra, 1991; Carter

et al., 2007; Wheeler et al., 2014; Wheeler and Kavanagh, 2017). Hotspots sourced from animal

carcasses, also referred to as carrion hotspots, significantly alter surface and belowground soil

physiochemistry and plant communities in terrestrial ecosystems (Carter et al., 2007; Keenan et

al., 2018). These alterations can have significant long-term impacts; for example large animal

carcasses had measurable effects on a prairie ecosystem for at least 5 years (Towne, 2000), and a

decade or more in the Arctic (Danell et al., 2002). In addition to providing a critical source of C

and N, carrion hotspots are important sources of ecosystem heterogeneity (Towne, 2000; Bump

et al., 2009b) and promote biodiversity (Barton et al., 2013).

      Carrion decomposition occurs in a series of physical (Payne, 1965) and biogeochemical

(Keenan et al., 2018) stages. The breakdown and release of animal tissues provides a labile

source of nutrients for insect and animal scavengers as well as soil microfauna (i.e., bacteria,

fungi, nematodes). Studies evaluating the consequences of carrion decay on soil biogeochemistry

have monitored decomposition on a range of timescales, from days (Metcalf et al., 2013;

Macdonald et al., 2014; Keenan et al., 2018; Szelecz et al., 2018) to years (Towne, 2000; Bump

et al., 2009a; Keenan et al., *in press*), and in different climatic and geographic settings, including

temperate forests (Melis et al., 2007; Cobaugh et al., 2015; Keenan et al., 2018) and Australian

rangeland (Macdonald et al., 2014), as well as under controlled laboratory settings (Carter et al.,



2008, 2010). Some of the key changes that occur in soils following the deposition and

decomposition of carrion include: changes to pH, electrical conductivity, oxygen availability, gas

fluxes ($CO_2$, $CH_4$, $N_2O$, $H_2S$), elevated rates of microbially-driven C and N cycling, and

dissolved compounds available to microbes ($NH_4^+$, $NO_3^-$, $Ca^{2+}$, $SO_4^{2-}$) (Melis et al., 2007;

Aitkenhead-Peterson et al., 2012; Keenan et al., 2018).

Many of the rapid, pulsed perturbations to soil C and N pools observed at carrion

hotspots, such as elevated microbial respiration rates (measured as $CO_2$ release) and changes to

soil pH, return to background biogeochemical conditions during the skeletal stage of

decomposition, when soft tissues have been largely or completely degraded by insect and animal

scavengers (Cobaugh et al., 2015; Keenan et al., 2018). However, certain biogeochemical

measures, including soil stable $\delta^{15}N$ composition, have been observed to remain enriched in soils

collected at carrion hotspots compared to background soils for a protracted period of time, up to

several years (Bump et al., 2009a ; Wheeler and Kavanagh, 2017). Soil stable isotopic

composition integrates all biogeochemical activity within the soil as well as inputs from plant or

animal matter. In contrast with $\delta^{15}N$ enrichment, no changes in soil $\delta^{13}C$ composition have been

observed in surface soils of decomposition hotspots (Wheeler and Kavanagh, 2017; Keenan et

al., 2018). A variety of studies have demonstrated the potential for natural abundances of $^{15}N$ to

be used as a tracer of ecological processes, including N input from animals (urea and feces) in N-

limited and isolated ecosystems (Erskine et al., 1998) and input of marine taxa (salmon

carcasses) to terrestrial and riparian areas (Kline et al., 1990; Koyama et al., 2005).

While $^{15}N$ enrichment due to carrion decomposition has been demonstrated in previous

work, these studies were limited to surface soils (maximum sampling depth of 10 cm) from the

center of the hotspots (Bump et al., 2009a; Wheeler and Kavanagh, 2017). This has left a gap in



our understanding of the spatial extent of carcass enrichment, which is ultimately necessary for

quantifying ecosystem impacts of these decomposition inputs. Given the potential for natural

abundance $^{15}$N to serve as a long-term tracer of decomposition processes, the goal of this study

was to evaluate spatial changes in stable $^{15}$N-enrichment at a carrion hotspot one year post-

decay. In particular, the lateral and vertical extent of stable isotope changes as a result of

enhanced biogeochemical reactions in a hotspot is largely unknown. Soils beneath and adjacent

to former carrion hotspots (up to ~40 cm, the extent of visible fluid migration) were expected to

remain $^{15}$N-enriched one year after decay. Additionally, isotopic enrichment was expected to

persist to at least 10 cm depth, the maximum depth examined in previous studies.

**2 Materials and Methods**

**2.1 Study area and sample collection**

The study site was a mixed deciduous forest in East Tennessee (36°0'1.0" N, 84°13'1.6"

W, ~330 m elevation). Soils were part of the Fullerton-Pailo complex and characterized as Typic

Paleudults (Soil Survey Staff, 2018). The A horizon extended to approximately 20 cm depth.

Five ~23 kg nuisance North American beaver (*Castor canadensis*) carcasses were placed frozen

within scavenger prevention enclosures (1.19 x 0.74 x 0.81 m) and allowed to decay naturally,

starting 31 July 2016. As part of a separate study, approximately 75 g of surface soil (0-5 cm

depth) was collected a total of five times during decay beneath each animal (Keenan et al.,

2018). For this study, soils were collected on 8 August 2017, one year after decomposition, and

after bones had been removed from the site. Soils were taken from surface transects as well as

from cores obtained at depth below three carrion hotspots. Approximately 30 g of soil from the

top 0-5 cm were collected using a 3 cm diameter auger within the hotspot—an area 80 cm in



diameter of visibly discolored soil. Surface samples were additionally collected along a linear

transect radiating from the hotspot at 20 cm intervals up to 140 cm (Fig. 1). Within the hotspots,

soils were cored to a depth of 50 cm using a 10 cm diameter auger; cores were partitioned into

depth intervals of 0-5, 5-10, 10-15, 15-20, 20-30, 30-40, and 40-50 cm depth (Fig. 1).

        All soil samples were homogenized to a uniform consistency in the field by hand

(changing nitrile gloves between samples), removing any rocks, roots, leaves, or vegetation

larger than 2 mm. Samples were transported to the lab and processed immediately. Aliquots were

oven-dried in triplicate at 105°C for 48 h to determine gravimetric moisture (Table 1, Table S1).

Once dried, subsamples were powdered in an agate mortar and pestle and stored in 1.5 mL tubes

until subsequent isotopic analyses. Samples (~25 mg) were transferred into 5 × 9 mm tin

capsules (Costech). Isotopic analyses were conducted at Washington University in St. Louis.

Samples, standards, and blanks were loaded into a Costech Zero Blank autosampler and

combusted in a Flash 2000 elemental analyzer. Soil $\delta^{13}C$ and $\delta^{15}N$ values were measured on a

Delta V Plus continuous-flow (Conflo IV, Thermo Fisher Scientific), isotope-ratio-mass

spectrometer. Standards included millet and acetanilide. Millet was used to evaluate linearity.

Sample carbon isotopic values were corrected for sample size and instrument drift using millet

and acetanilide, and nitrogen values were corrected using millet, acetanilide, and urea. Analytic

precision was <0.2 ‰ for both carbon and nitrogen. Results are presented in δ notation as parts

per mil (‰) where $\delta^{13}C = [((^{13}C/^{12}C_{sample} / {}^{13}C/^{12}C_{standard}) - 1) \times 1,000]$ and $\delta^{15}N = [((^{15}N/^{14}N_{sample}$

$/ {}^{15}N/^{14}N_{standard}) - 1) \times 1,000]$. Vienna Pee Dee Belemnite was used as the carbon standard and

air was used for nitrogen.

        Soils collected from the center (0-5 cm depth) of all five hotspots and a composite control

sample (pooled soil collected from five locations ~3 m from each hotspot) were also analyzed for



microbial respiration rates (as evolved $CO_2$), ammonium, nitrate, nitrification potential, pH, electrical conductivity, dissolved organic C and N, and protein content, building from a previous study at the site and following the methods described by Keenan et al. (2018). In brief, headspace $CO_2$ was measured immediately after placing and sealing soil into 60 mL serum vials,

as well as after 24 h (LI-820, Licor Inc.). Vacuum-filtered (1 μm; Ahlstrom, glass microfiber) soil extracts (10 g soil: 40 mL 0.5 M $K_2SO_4$) were collected after shaking for 4 hours at 150 rpm at room temperature, and were frozen at -20°C until subsequent colorimetric analysis of ammonium and nitrate (Rhine et al., 1998; Doane and Horwath, 2003). Aliquots were oxidized with a persulfate solution to measure dissolved organic carbon (DOC) as evolved $CO_2$ and

dissolved organic nitrogen (DON) colorimetrically as nitrate (Doyle et al., 2004). Nitrification potential was determined colorimetrically using a modified chlorate block method optimized for microplates (Belser and Mays, 1980; Keeney and Nelson, 1982; Hart, 1994). Soil pH and electrical conductivity were measured from a soil slurry (3 g soil: 6 mL deionized water) using a handheld multi-parameter meter (Orion A329, Thermo Scientific). Protein content was

determined using the Bradford Assay (Wright and Upadhyaya, 1996; Redmile-Gordon et al., 2013). Because the goal of this study was to focus specifically on stable isotopes as long-term tracers in carrion hotspots, only surface soils from the five remnant hotspot centers were processed for full physiochemistry.

**2.2 Stable isotope analyses**

        The contribution of carcass-derived nitrogen to bulk soil stable isotopic composition in surface transects was determined using a linear two-member isotope mixing model (Wheeler and Kavanagh, 2017; Keenan et al., 2018), using bulk control soil $\delta^{15}N$ composition (0.1 ‰) as one



end-member and beaver decomposition fluid (10.2 ‰) as the other. Decomposition fluid is the

by-product of microbial and autolytic processes acting on a carcass after animal death. Fluids

consist of amino acids, dead and live microbial cells, urea, water, and lipids, and represent one of

the primary mechanisms for return of host's tissues to the surrounding environment.

Decomposition fluid isotopic composition was previously determined, using fluids collected

from three decomposing beavers left on a shallow plastic tray to intercept fluids (Keenan et al.,

2018). The linear equation for the isotope mixing model (Wheeler and Kavanagh, 2017) was

defined as:

$$CDN = [(TEM - SEM)/(FEM - SEM)] \times 100$$

Where CDN is the carcass-derived N (%), TEM is the average $\delta^{15}$N of soil from the treatment

condition (sampling interval along the surface transects), SEM is the end-member control soil

stable isotopic composition (0.1 ‰), and FEM is the end-member isotopic composition of

decomposition fluids (10.2 ‰). The contribution of control soil-derived $\delta^{15}$N to measured

treatment conditions was calculated by subtracting CDN (%) from 100 %.

        The contributions of multiple sources to bulk soil stable isotopic composition ($\delta^{13}$C and

$\delta^{15}$N) were assessed using Stable Isotope Analysis in R (SIAR) using the simmr package (Parnell

et al., 2010). Three sources were integrated into the modeling: (1) surface control soils, (2)

decomposition fluids, and (3) control soils at depth (40 cm). Soil $\delta^{13}$C and $\delta^{15}$N composition at

each depth were modeled to assess which source exerted a greater influence on bulk soil isotopic

composition.

        To track changes in $\delta^{15}$N between surface soil and soil collected at depth, $\Delta^{15}$N values

were calculated by subtracting the $\delta^{15}$N value of soil at each depth from values obtained at the





surface of the hotspot and control sampling locations. Negative $\Delta^{15}$N values indicate that surface

soils are $^{15}$N-enriched compared to soils at depth (Martinelli et al., 1999).

### 2.3 Statistical analyses

Data were analyzed using SigmaPlot to test for significant differences between treatments

and controls. For both surface and depth transects, data from the three transects were treated as

replicates for subsequent statistical analyses. Significance ($p < 0.05$) was determined based on

one-way ANOVA analyses with Holm-Sidak post-hoc testing. Significant differences between

control and hotspot soils were determined using paired t-tests at each sampling depth or transect

interval using R (R Core Team, version 3.5.0).


### 3 Results

### 3.1 Surface soil biogeochemical changes during decomposition

During carrion decomposition, fluids sourced from the carcass were released into the

surrounding environment (Fig. 2). The pulse of nutrient-rich fluids resulted in significant

changes to surrounding soil physiochemistry (Table 1, Table S1). Soils exhibited long-term

changes to physiochemistry following fluid degradation by soil microbial communities. In

particular, after one year of decay, soil pH was significantly lower than control, initial, and pre-

decay soils ($p < 0.001$; $F = 59.317$). In addition, bulk soil $\delta^{15}$N remained significantly enriched

compared to control and starting soil isotopic composition ($p < 0.001$; $F = 27.948$). Other

physicochemical parameters, including conductivity, microbial respiration, DOC, DON,

ammonium, nitrate, and nitrification potential all returned to background conditions after one





year. With the exception of the one year samples, data were previously published in Keenan et al. (2018) and are included here for comparison.

### 3.2 Lateral changes in stable isotopic composition


Soils were significantly [15]N-enriched within the visible carrion hotspot (mean soil composition $7.5 \pm 1.0$ ‰) and up to 60 cm from the hotspot center ($2.2 \pm 0.5$ ‰) compared to composite control soils (0.1 ‰) (Fig. 3, Table S2) (paired t-test, p = 0.016). Soil $\delta^{15}N$ values gradually declined, reaching background abundance values around 80 cm of the hotspot center. In contrast, there were no significant differences between control and hotspot soil $\delta^{13}C$, and no differences as a function of distance from the hotspot center (one-way ANOVA, p = 0.464; F = 1.004). C/N ratios were lower within the hotspot and exhibited a gradual and significant increase with increasing distance from the hotspot center (Fig. 3). However, there were no significant differences between hotspot and control soil C/N.


The influence of carrion decomposition on soil stable $\delta^{15}N$ isotopic composition decreases with increasing distance from the hotspot (Figs. 3a, S1). Soil C:N composition follows a linear trend, increasing by 0.07 per cm from the hotspot center (Fig. S1). Based on linear two-member isotope mixing models, carcass-derived fluids exhibit a linear decrease in contributions to soil isotopic composition with increasing distance from the hotspot. Carcasses contribute to soil stable $\delta^{15}N$ isotopic composition up to 60 cm from the hotspot center (Fig. 4), an area that was not visibly discolored as observed within the center of the hotspot (Fig. 2).



### 3.3 Vertical changes in stable isotopic composition



Soil collected at depth beneath the three mortality hotspots was significantly $^{15}$N-enriched

compared to control soils up to 10 cm depth (Fig. 5). Surface hotspot soils were also enriched at

30 cm depth compared to the control. Control soils became more $^{15}$N-enriched with increasing

depth. There was no significant difference between control and hotspot soil $\delta^{13}$C and C/N values,

and both exhibited the same trends with depth. Soils exhibited $^{13}$C-enrichment with increasing

depth and a decline in C/N ratios.

Control soils exhibited a strong positive linear relationship between the negative of the

natural log of bulk soil %N and %N and stable isotopic composition, reflecting decreasing N and

C availability with increasing depth. Decomposition results in a shift in the hotspot soil N

isotopic discrimination factor ($D$, or the slope of the linear regressions) (Natelhoffer and Fry,

1988), leading to a less positive slope compared to control soils (Fig. 6). $D$ does not change for C

isotopes in control or hotspot soils.

In general, soils exhibit a trend of increasing $\Delta^{15}$N (the difference between soil $\delta^{15}$N

value at a specific depth and $\delta^{15}$N at the surface) with depth, reflecting $^{15}$N-enrichment in deep

forest soils (Martinelli et al., 1999). Hotspot soils exhibited lower $\Delta^{15}$N values compared to

control soils, indicating little change in $^{15}$N-enrichment with depth (Table 2). Control soils

displayed increasing $\Delta^{15}$N with depth, a pattern globally observed in forest soils (Martinelli et al.,

1999) (Table 2). Combined with $D$ (Fig. 6), hotspot and control vertical profiles have distinct N

sources and exhibit distinct isotopic enrichment patterns with depth.

Stable isotope mixing models evaluated the proportional contribution of three distinct

sources to soil stable isotopic composition in hotspot depth profiles: control surface soils,

decomposition fluids, and control soils at 40 cm depth. Because the lateral influence of

decomposition on soil $\delta^{15}$N composition did not extend beyond 60 cm in surface soils, samples





collected at 100, 120, and 140 cm were included in the control surface soil average  (-0.1 ± 0.3 ‰). Soils collected from 0-5 cm exhibited a significant contribution from decomposition fluids (Fig. 7a,b). As depth increases beyond 10 cm, there is no change to the proportional contribution

of decomposition fluid to bulk stable isotopic composition. By 30-40 and 40-50 cm depth, hotspot soil $\delta^{15}$N and $\delta^{13}$C compositions are similar to control soils (Fig. 7a), indicating limited, if any, input from decomposition fluids.

## 4 Discussion

Soils associated with carrion decomposition hotspots retained biogeochemical markers of vertebrate decay at least one year after soft tissue degradation. Within the hotspots, soils remained $^{15}$N-enriched compared to control locations, suggesting that decomposing animals have the potential to exert long-term changes, here at least one year, on surface and subsurface soil stable isotopic composition. The beaver carcasses used this study, which were between 20 kg and

25 kg in mass, resulted in measurable changes down to 10 cm depth and out to 60 cm away from the center of the hotspot, beyond the area that was visibly discolored from decomposition fluids. The contribution of carcass-derived N to soils at depth as well as laterally is influenced by a variety of physical and climatic variables. Here, decay occurred during the summer in East Tennessee, with an average high of 32.2°C for the month of August. Carcasses were exposed to

measureable precipitation six out of 10 days after placement, preventing soft tissues from significant desiccation and supporting abundant blowfly larvae and other insect activity. Blowfly larvae migrating away from the carcasses on the surface and within the soil to pupate likely provided an important physical mechanism to distribute beaver-enriched N to surrounding soils. Blowfly larvae can potentially move up to 10 m away from the carcass, and typically extend



down into the soil up to 10 cm depth, depending on the soil substrate properties (Gomes et al.,

2006). As blowfly larvae disperse, they have the potential to physically transport decomposition

fluids acquired internally or externally during feeding, release excrement during migration, or

die, leaving their tissues to degrade. An estimated 65.9 % of pupae that disperse to pupate die en

route (Putman, 1977). Rainfall may have also contributed to the downward movement of

decomposition fluids.

      The temporal persistence of isotopic enrichment hotspots is currently unknown, but is

likely to be ecosystem, carrion type, and carrion mass-specific. A larger carcass would be

expected to result in greater lateral and vertical dispersal of carrion-derived fluids, as well as

greater changes to ecosystem processes, because of the greater volume of decomposing soft

tissue (Baruzzi et al., 2018). In addition, larger carcasses may host a larger and longer-lived

insect community (Parmenter and MacMahon, 2009), including blowfly larvae, which may

further nutrient dispersal and may impact a larger area. Bump et al. (2009b) observed elevated

foliar $\delta^{15}N$ values in plants growing on sites impacted by deer carcass (~56 kg) decomposition at

least 2.5 years after decay in a temperate hardwood forest, suggesting a long-lived hotspot

signature. In some ecosystems, such as the Arctic tundra, isotopic enrichment is likely to persist

for even longer based on perturbations to C and N surrounding muskox after 5 to 10 years of

decay (Danell et al., 2002).

      Increasing $\delta^{13}C$ and $\delta^{15}N$ values with depth in soils has previously been observed in a

variety of soil types and climatic conditions (Natelhoffer and Fry, 1988; Martinelli et al., 1999;

Billings and Richter, 2006). Changes to $\delta^{13}C$ with depth are due to progressive cycling of C

through microbial biomass (Liang et al., 2017), where selective preservation and biochemical

fractionation together lead to $^{13}C$-enriched organic C in soil (Natelhoffer and Fry, 1988; Billings



and Richter, 2006). While we observed a similar increase in $\delta^{13}C$ with depth, we did not see a

significant change in $^{13}C$ as a result of carcass enrichment. Wheeler and Kavanagh (2017)

similarly did not observe a change in soil $\delta^{13}C$ following carrion decomposition.

Increasing $\delta^{15}N$ values with depth reflects two broad biochemical processes leading to

fractionation, both likely driven by microbial activities. First, the preferential excretion of $^{15}N$-

depleted compounds during catabolism and anabolism leaves the residual microbial cells and soil

$^{15}N$-enriched. Second, kinetic fractionation associated with gaseous N loss is also known to

result in enrichment, depending on the microbial communities present and N mineralization rates

(Evans, 2001; Robinson, 2001; Liang et al., 2017). Over time, as soil profiles develop, accretion

of $^{15}N$-enriched microbial cells, particularly fungi, leads to isotopic enrichment at depth (Billings

and Richter, 2006). In contrast, plant and leaf litter are the dominant contributors to N pools in

surface soils in most temperate forest ecosystems (Vidon et al., 2010), resulting in surface soils

that are isotopically-depleted compared to the soil profile at depth. Decomposition hotspots,

however, disrupt the expected pattern (Fig. 5), causing surface enrichment, and likely leave a

lasting impact on soil stable isotopic composition.

For systems at or near steady state conditions, the difference in isotopic enrichment

between soils at depth and the surface ($\Delta^{15}N$) provides a way to compare soils from different

geographic and climatic locations (Martinelli et al., 1999), and was used here to compare hotspot

soils and those collected at control locations. $\Delta^{15}N$ values observed in the control depth profile

are within the expected range observed in temperate forests worldwide (2.7 to 9.1 ‰) (Table 2).

However, as a consequence of carrion inputs and decay, $\Delta^{15}N$ values are more similar to those

observed in tropical forest ecosystems (1.1 to 4.3 ‰). In tropical systems, lower $\Delta^{15}N$ values are

thought to reflect more open N cycling with elevated N losses (nitrification, nitrate leaching, and



ammonia volatilization) under conditions of elevated total N inputs (Martinelli et al., 1999). Whether our observed changes in $\Delta^{15}N$ are due to elevated N cycling rates, disequilibrium effects across the soil profile due to changing N inputs from a system dominated by atmospheric dry and wet deposition of nitrate and ammonium to one with carrion-sourced N, or both, is not known.

The stable isotopic discrimination factors ($D$) did not differ for control and hotspot soil $\delta^{13}C$ (Fig. 6), suggesting that C cycling and pools in soils one year after carrion decay are not altered. In contrast, $D$ values for $\delta^{15}N$ were different between control and hotspot soils. This indicates distinct N sources for the two soil profiles, emphasizing that decaying carrion provide an important and potentially distinct N pool for soil ecosystems. In addition, differences in $D$

values between the two soils suggest that there is less discrimination occurring within hotspot soils compared to control soils, likely due to the rapid input of an isotopically-enriched N pool (Evans, 2001).

Hotspot soils received the input of beaver-derived fluids (10.2 ± 0.4 ‰) (Keenan et al., 2018) as well as soft and hard tissues (1.0 to 4.0 ‰ for beaver bone collagen from Minnesota;

Fox-Dobbs et al., 2007). Stable isotope mixing models demonstrate that decomposition fluids are a significant contributing source to bulk soil stable isotopic composition up to 60 cm from the hotspot center (Fig. 4), and multi-source mixing models also suggest some contributions up to 10 cm depth (Fig. 7). Beyond 10 cm depth, there is a stable and uniform proportional contribution, with decomposition fluids not significantly influencing soil stable isotopic composition. Rather,

natural $\delta^{15}N$ enrichment due to soil accretion processes can explain the observed soil stable isotopic composition. Based on the isotopic discrimination factor ($D$) for N in hotspot soils, linear regression line for hotspots soil results in a predicted starting N source of 4.8 ‰, in line with the expected value for fresh beaver tissues (Fig. 6). This suggests that the stable isotopic

composition of soil δ$^{15}$N, even one year after decay, may be a useful tool to infer starting animal

tissue isotopic composition.

## 5 Conclusions

Soil isotopic composition is controlled by the isotopic composition of input(s) and

subsequent abiotic and biotic biogeochemical processes that potentially lead to fractionation. The

input of an isotopically-enriched source such as decomposing carrion resulted in significant

spatial and temporal perturbations to soil stable isotopic composition. The decay of ~23 kg North

American beavers resulted in rapid (within days) and long-lived (up to one year) $^{15}$N-enrichment

in forest soils up to 10 cm depth and ~60 cm distal. Soil biogeochemistry, particularly N cycling,

is complex, and carrion inputs have the potential to alter expected patterns long after soft tissues

have been completely degraded.

Observed $^{15}$N-enrichment at depth and laterally is likely due to a combination of physical

movement of fluids during decomposition and the transport of fluids by insects, particularly

blowfly larvae. In this system, rainfall during decomposition may have also acted as a physical

transport mechanism. While likely to be significantly influenced by carcass size, climate, and

soil type, decomposition has the potential to exert long-lived influences on soil stable isotopic

composition.

*Data availability*

All data generated in this study are available in the Supplement.

*Author contributions.* SWK and JMD designed the experiments. All authors assisted with data

interpretation. JMD and SMS provided financial, lab, and analytical resources. SWK and JMD

prepared the manuscript with contributions from SMS.

*Competing interests.* The authors declare that they have no conflict of interest.


*Acknowledgements.* Salvaged nuisance beavers were provided by the USDA, APHIS, Wildlife

Services of East Tennessee. Lois S. Taylor, Jose Liquet, Fei Yao, and Jialin Hu assisted with soil

collection. Alex Bradley provided lab and instrument access, and Melanie Seuss assisted with

stable isotopic analyses. This research was funded by a National Science Foundation Award

(EAG1549726) to JMD and SMS.

### *Figure and table legends*

Figure. 1: Schematic cross-section view of the locations of soil samples (stars) collected from

each of three carrion decomposition sites. Dashed line represents the hotspot—the area of visibly

discolored soil. Soils collected at depth extended to the B horizon. The visibly discolored area of

soil due to carrion hotspot formation extended approximately 35-40 cm from the hotspot center

along the surface and to a few centimeters depth.

Figure 2: View of a beaver after placement (a) and during advanced decay (b) to demonstrate the

lateral migration of carcass-derived fluids during decay. Both photos are from the same animal,

and (b) were taken during advanced decay (8 August 2016). Visible extent of fluid migration is

outlined in the white dashed line.





Figure 3: Lateral changes in soil stable (a) $\delta^{15}$N and (b) $\delta^{13}$C isotopic composition and (c) C/N

ratios extending from carrion hotspot centers. Soil was visibly discolored 35-40 cm from the

center (here, 0 cm distance). Letters indicate significant differences as a function of distance

based on an ANOVA and Holm-Sidak post-hoc test, and asterisks denote significant differences

between control and hotspot soils (t-test). The dashed line represents control surface soil (0-5

cm) composition.


Figure 4: Results of linear two-member isotope mixing distinguishing the contributions of soil

and carcass fluid to bulk soil stable isotopic composition.

Figure 5: Stable isotopic composition and C/N ratios for soils beneath carrion hotspots (closed

circles) and at a control location (stars). Letters indicate significant differences as a function of

depth, and asterisks indicate significant differences between control and hotspot soils (both based

on one-way ANOVA, $p < 0.05$).

Figure 6: Bulk soil stable isotopic composition and corresponding negative natural log %N (a)

and %C (b) for hotspot and control soils with depth. Linear regressions were fit to hotspot and

control datasets.

Figure 7: Stable isotope mixing models for hotspot soils collected at depth. (a) The three

potential sources contributing to soil $\delta^{13}$C and $\delta^{15}$N isotopic composition were used as inputs,



and exert different contributions at each depth. (b) The proportional contribution of

decomposition fluids changes as a function of depth.

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



Table 1: Selected soil biogeochemical data during one year of decomposition. Letters indicate significant differences between samples based on One-way ANOVA (p < 0.05). Asterisks indicate significant differences between control and treatment soils.

| | Sampling Date | Soil Gravimetric Moisture | pH | Conductivity (µS cm⁻¹) | Dissolved Oxygen (%) | Total carbon (%) | Total nitrogen (%) | C/N | δ¹⁵N | δ¹³C |
|---|---|---|---|---|---|---|---|---|---|---|
| *Initial* | 29 July | 0.299 ± 0.012^AB | 6.79 ± 0.1^A | 47.83 ± 5.9^AC | 98.5 ± 0.29^A | 5.11 ± 0.101 | 0.295 ± 0.009 | 17.33 ± 0.25^AC | 1.48 ± 0.23^A | -27.86 ± 0.08 |
| *Early* | 1 August | 0.216 ± 0.034^A | 6.86 ± 0.3^A | 73.53 ± 32.2^A | N.M. | 3.927 ± 1.08 | 0.245 ± 0.058 | 15.93 ± 1.7^AC | 2.65 ± 1.66^A | -27.73 ± 0.42 |
| Early control | 1 August | 0.219 | 6.82 | 36.78 | N.M. | 4.196 | 0.247 | 16.99 | 1.76 | -27.56 |
| *Active* | 3 August | 0.234 ± 0.056^A | 8.64 ± 0.3^B | 2150.48 ± 1282^BC | 9.16 ± 7.89^BC | 4.251 ± 0.798 | 0.362 ± 0.096 | 12.01 ± 1.34^BC | 6.23 ± 1.50^B | -27.77 ± 0.30 |
| Active control | 3 August | 0.160 | 6.68* | 31.65* | 98.6 ± 1.25* | 4.159 | 0.267 | 15.58* | 1.48* | -27.68 |
| *Advanced* | 9 August | 0.286 ± 0.081^AB | 8.78 ± 0.1^B | 1233 ± 494^C | 19.4 ± 31.2^B | 4.000 ± 1.29 | 0.303 ± 0.080 | 13.07 ± 0.75^BC | 8.72 ± 2.09^B | -27.64 ± 0.20 |
| Advanced control | 9 August | 0.223 | 6.84* | 43.42* | 98.0 ± 0.57* | 5.008 | 0.281 | 17.82* | 1.26* | -27.77 |
| *Early skeletal* | 6 September | 0.242 ± 0.070^A | 7.58 ± 0.4^C | 973.8 ± 211^AC | 97.4 ± 0.84^AC | 3.610 ± 0.839 | 0.293 ± 0.057 | 12.28 ± 0.74^BC | 9.26 ± 1.54^B | -27.74 ± 0.30 |
| Early skeletal control | 6 September | 0.121 | 6.84* | 35.08 | 98.3 ± 0.50 | 4.023 | 0.259 | 15.53* | 1.78* | -27.63 |
| *Late skeletal* | 9 December | 0.271 ± 0.021^AB | 6.93 ± 0.3^A | 225.2 ± 84.8^AC | 100 ± 0^A | 2.668 ± 0.352 | 0.214 ± 0.030 | 12.51 ± 0.67^BC | 9.25 ± 1.33^B | -27.41 ± 0.25 |
| *Late skeletal conrol* | 9 December | 0.246 | 6.73 | 29.13 | 100 ± 0 | 2.084 | 0.136 | 15.37* | 1.79* | -27.30 |
| ***1 yr. post decay*** | **10 August** | **0.404 ± 0.027^B** | **6.10 ± 0.3^D** | **29.47 ± 7.6^A** | **N.M.** | **4.253 ± 1.07** | **0.285 ± 0.036** | **15.24 ± 3.49^C** | **8.42 ± 1.52^B** | **-27.67 ± 0.25** |
| ***1 yr. post decay control*** | **10 August** | **0.449*** | **6.29*** | **23.07** | **N.M.** | **4.36** | **0.26** | **17.08** | **0.05*** | **-27.73** |





Table 1 (continued):

| | Protein (mg g$^{-1}$) | Microbial respiration rate (µg CO$_2$-C release gdw$^{-1}$ day$^{-1}$) | DOC (µg C gdw$^{-1}$) | Ammonium (mg NH$_4$-N gdw$^{-1}$) | Nitrification potential rate (mg NO$_2$ gdw$^{-1}$ day$^{-1}$) | Nitrate (mg NO$_3$-N gdw$^{-1}$) | DON (mg N gdw$^{-1}$) | Accumulated Degree Days (ADD) |
|---|---|---|---|---|---|---|---|---|
| *Initial* | 0.251 ± 0.051 | 50.9 ± 8$^A$ | 2.44 ± 0.2$^{AC}$ | 0.039 ± 0.01$^A$ | 0.181 ± 0.05$^A$ | 0.000 ± 0.0$^A$ | 0.283 ± 0.02$^A$ | 26.1 |
| *Early* | 0.200 ± 0.037 | 51.4 ± 18$^A$ | 3.44 ± 1.0$^A$ | 0.101 ± 0.06$^A$ | 0.237 ± 0.16$^A$ | 0.000 ± 0.0$^A$ | 0.718 ± 0.35$^A$ | 106.7 |
| Early control | 0.159 ± 0.004 | 35.2 | 2.35 | 0.011 | 0.167 | 0.000 | 0.242 | 106.7 |
| *Active* | 0.260 ± 0.025 | 300 ± 90$^B$ | 66.54 ± 40.4$^B$ | 2.49 ± 1.24$^B$ | 0.366 ± 0.09$^A$ | 0.001 ± 0.0$^A$ | 1.363 ± 1.4$^A$ | 160.3 |
| Active control | 0.225 ± 0.007 | 27.8* | 2.52* | 0.010* | 0.163 | 0.000 | 0.205 | 160.3 |
| *Advanced* | 0.281 ± 0.072 | 162.8 ± 110$^A$ | 42.5 ± 30.0$^C$ | 2.29 ± 1.80$^{BC}$ | 0.517 ± 0.17$^A$ | 0.001 ± 0.0$^A$ | 1.906 ± 1.7$^A$ | 321.7 |
| Advanced control | 0.239 ± 0.024* | 45.7* | 3.44* | 0.015* | 0.181 | 0.003 | 0.304 | 321.7 |
| *Early skeletal* | 0.238 ± 0.049 | 76.9 ± 42$^A$ | 14.3 ± 3.4$^{AC}$ | 0.775 ± 0.14$^{AC}$ | 8.57 ± 4.4$^B$ | 0.309 ± 0.169$^B$ | 6.089 ± 1.24$^B$ | 1042.8 |
| Early skeletal control | 0.193 ± 0.021 | 20.8 | 2.89 | 0.006 | 0.130* | 0.000* | 0.185* | 1042.8 |
| *Late skeletal* | 0.250 ± 0.014 | 57.2 ± 26$^A$ | 10.2 ± 8.4$^{AC}$ | 0.246 ± 0.06$^A$ | 0.017 ± 0.20$^A$ | 0.019 ± 0.001$^A$ | 1.929 ± 0.37$^A$ | 2591.7 |
| *Late skeletal conrol* | 0.195 ± 0.016 | 46.5 | 3.12 | 0.012 | 0.039 | 0.000 | 0.309 | 2591.7 |
| ***1 yr. post decay*** | **0.239 ± 0.021** | **64.1 ± 8$^A$** | **2.81 ± 0.5$^A$** | **0.008 ± 0.0$^A$** | **0.006 ± 0.00$^A$** | **0.001 ± 0.0$^A$** | **0.095 ± 0.01$^A$** | **6377.5** |
| ***1 yr. post decay control*** | **0.195 ± 0.008** | **59.5** | **2.43 ± 0.0** | **0.007** | **0.006** | **0.000** | **0.093** | **6377.5** |



Table 2: Differences in soil $\delta^{15}N$ at depth and $\delta^{15}N$ in surface soils for hotspot and control depth

profiles.

| Depth (cm) | $\Delta^{15}N$ (‰) | |
| --- | --- | --- |
| | Hotspot | Control |
| 0 | 0 | 0 |
| 5 | -2.1 | 2.65 |
| 10 | -1.5 | 3.2 |
| 15 | 0.1 | 6.6 |
| 20 | 0.9 | 7.7 |
| 30 | 0.9 | 6.1 |
| 40 | 1.2 | 8.4 |






# Figure 1

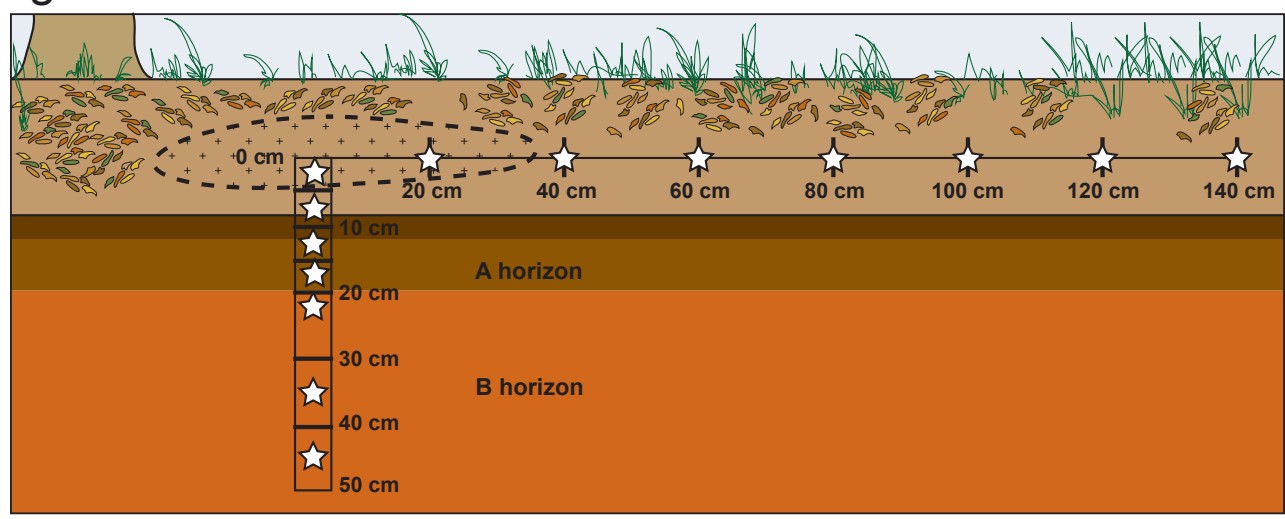



## Figure 2





## Figure 3



# Figure 4

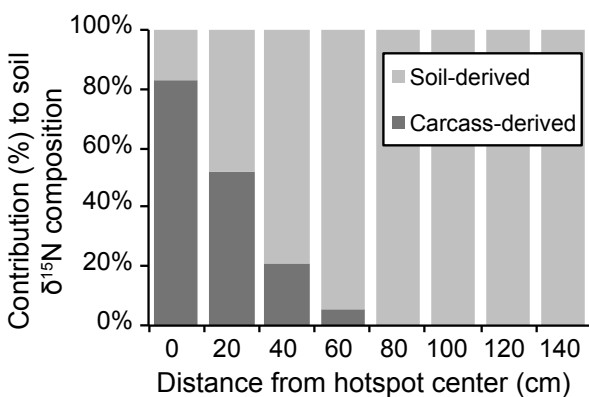



Figure 5

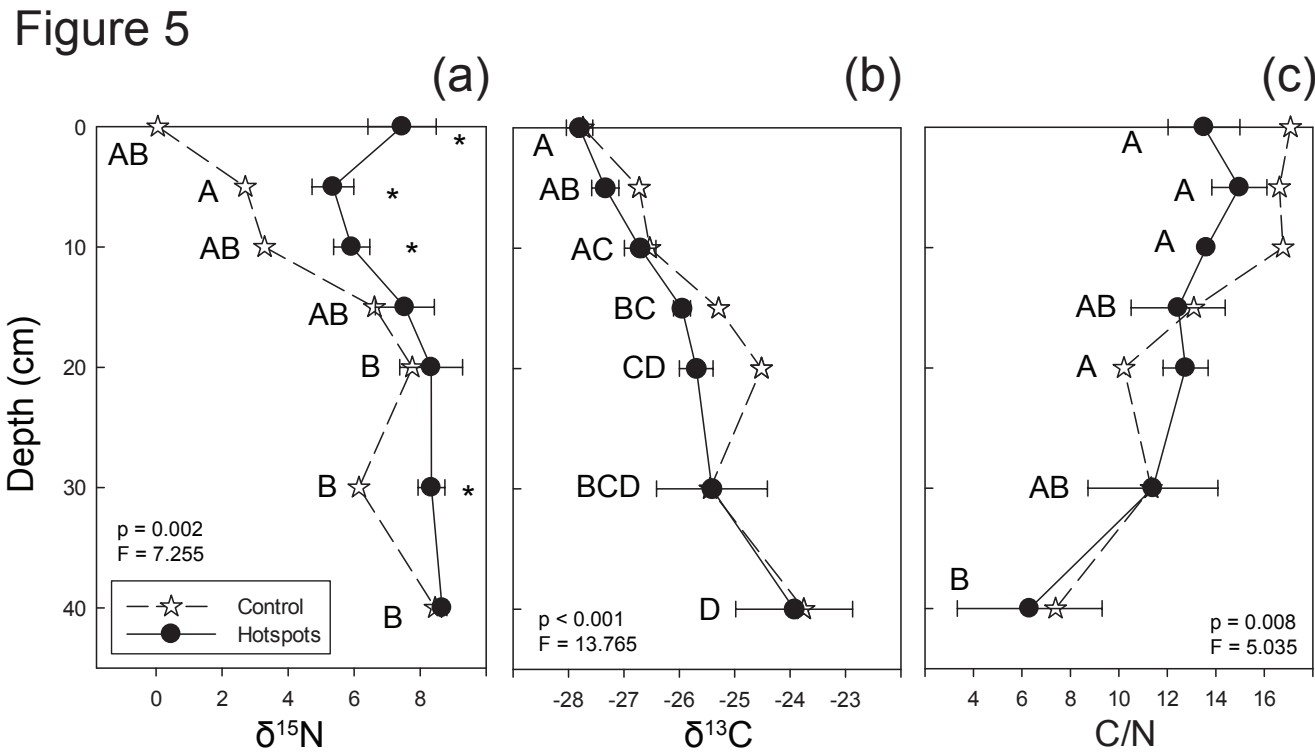



## Figure 6

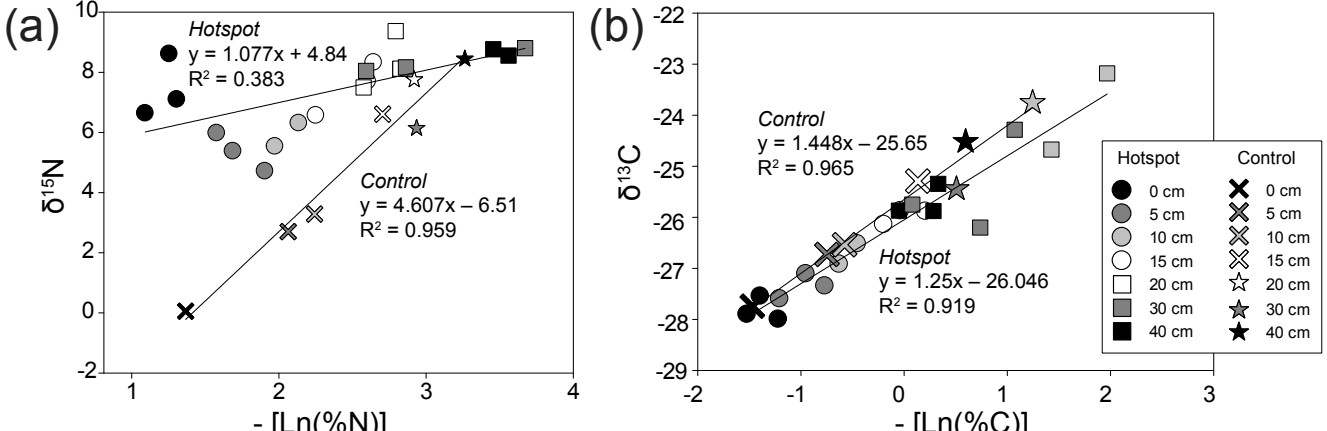




# Figure 7

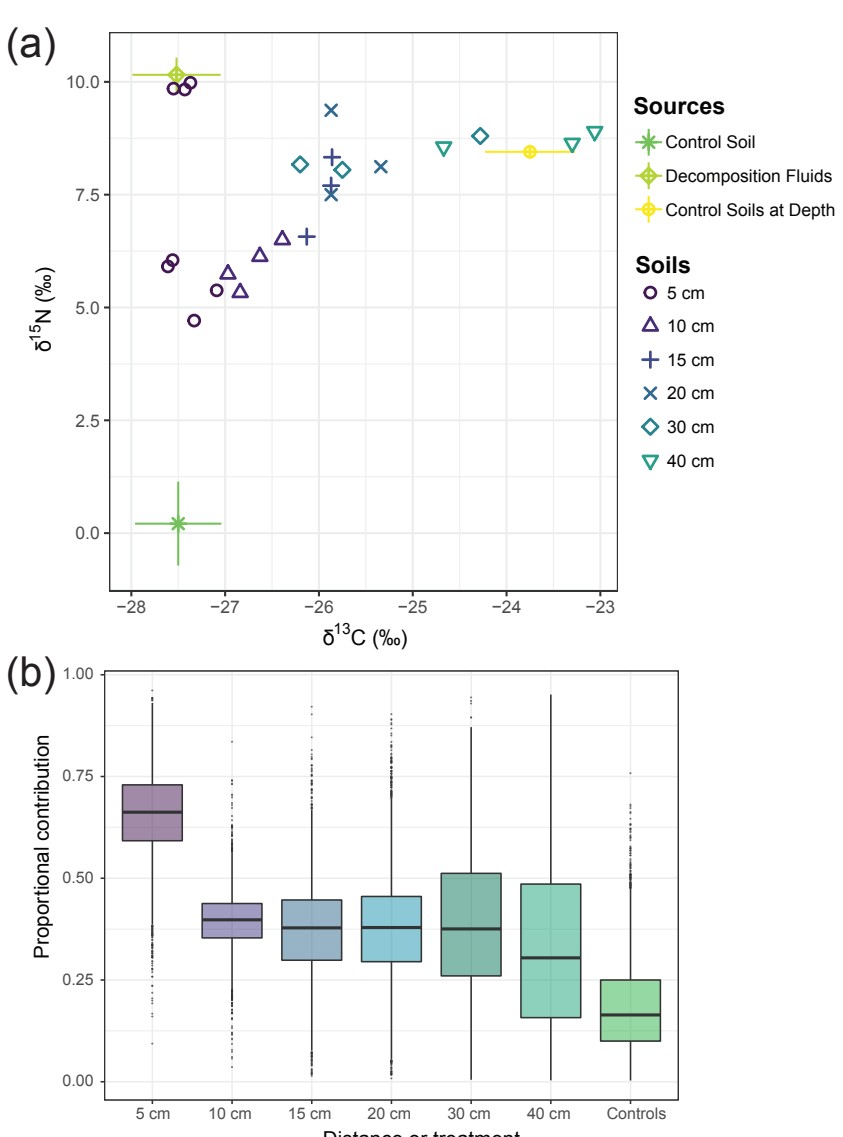