# Peer review of "Spatial changes in soil stable isotopic composition in response to carrion decomposition"

_Biogeosciences, 2018_

## Referee Comment (RC1) · Anonymous Referee #1 · 23 Jan 2019

Overall the manuscript, "Spatial changes in soil stable isotopic composition in response to carrion decomposition," within minimal revision is a well written and a sound contribution towards understanding the spatial influence of pulsed organic nutrient inputs into terrestrial ecosystems from the deposition of carrion. Numerous studies have approached this subject but the geographic expanse and complexity of the resultant biogeochemical responses leaves ample room for investigation. This work helps to bridge the gap between previous studies through both the spatial layout of the observations and the utilization of isotope discrimination factors and $\delta15N$ methodologies to tease apart the spatial extent of carrion influence within the soil profile.

Addressing the following concerns and comments will enhance the quality of this manuscript:

[Figure]

Specific comments:

Table 1: Insert note that defines N.M.

Perhaps note that control values do not have an error term due to being homogenized into a single sample.

The table caption or a note should include the statement about this data being from Keenan et al. 2018 except for the one year data. Also include a reprint permission statement in text (section 180) and with the table if required by either journal.

The caption states that the letters indicate differences between samples but it is not clear as to between which samples the letters are referring to from the caption or from what was readily found in the text.

Figure 3. Again the caption states that the letters indicate differences between samples but it is not clear as to between which samples the letters are referring to from the caption or from what was readily found in the text.

Figure 5. Similar comment to Figure 3. Clearly there are differences signified with depth but it is not readily apparent what the difference is between A, AB, B, etc.

Technical corrections:

Section 110: The equations as written may prove confusing to readers unfamiliar with isotope ratio calculations due to the use of the backslash as the division symbol both within the numerator and denominator as well as between. Perhaps something like 13C/12Csample $\div$ 13C/12Cstandard would be better.

Section 175: The following sentences seem to be restating a similar conclusion, "The pulse of nutrient-rich fluids resulted in significant changes to surrounding soil physio-chemistry (Table 1, Table S1). Soils exhibited long-term changes to physiochemistry following fluid degradation by soil microbial communities." Consider strengthening this paragraph by combining or differentiating these statements.

Section 185: ". . . values around 80 cm of the hotspot" presumably should read "80cm from the hotspot".

Section 200: Finesse this sentence a little bit to clarify that the 60 cm extent was beyond the carcass decomposition island. I believe that is what you are trying to state.

Section 235: The flow and the strength of the second sentence could be enhanced by revising the inclusion of ", here at least one year,". This is an important point that should specifically state that the results are for the given location, climate, soil, etc. and perhaps it would be better to give this its own subsequent sentence.

Typo - The third sentence, "The beaver carcasses used this study," should be "used in this study".

Section 315: Typo - Sentence missing "a", "Based on the isotopic discrimination factor (D) for N in hotspot soils, a linear regression . . . . ."

---

## Referee Comment (RC2) · Lukas Kohl (Referee) · 16 Jul 2019

Lukas Kohl (Referee)

lukas.kohl@mun.ca

Review Keenan et al, Biogeosciences July 16th, 2019

General comments

Keenen and co-authors investigated the effect of carrion decomposition on the underlying soil. In particular, they studied the spatial extent to a beaver carrion decomposition hotspot changed soil biogeochemical parameters (mainly C:N and d15N) one year post deposition. They find that elevated d15N values due to N inputs from the decomposing beaver were detected to 60cm lateral and 10cm depth.

The manuscript covers an important and understudied topic of terrestrial ecosystem ecology. The authors used state of the art methods and their results justify their conclusions. The manuscript reads very nicely and is surely of high interest to the Biogeosciences readership.

Specifc comments

- I think the main weakness of the manuscript is that the authors pooled all control samples (soils collected in some distance from the placed beavers) and analysed only a single composite sample. This means we cannot know the spatial variability of control soil properties, or the uncertainties associated with the measured average.

- The manuscript's use of biogeochemistry is somewhat confusing (e.g. L19-21). In my opinion, changes in soil d15N values may result from either changes in soil N biochemistry, or from changes in the d15N values of N inputs to soils. The manuscript's data largely suggest the latter is the dominant effect observed here. Where actual changes in the soil biogeochemistry are implied (again, e.g. L19-21), it would be better to be more specific and describe the changes in soil biogeochemistry that they think are indicated by changes

- I think that assumptions that are needed for the 13C/15N three-endmember mixing model to calculate input sources for deeper soil layers are likely not met. Such a model assumes that C and N of a given soil sample originate in the same proportions from the same sources, which is not true .

Furthermore, the authors need to clarify what the mixing model acutally estimates (e.g. L223: ".. evaluated the proportional contributions of three distinct sources to the stable isotopic composition in hotspot deep profiles ..") - mixing models do not estimate contributions to the isotopic composition, but to the contribution of distinct sources to a particular pool of matter (soil organic matter, soil nitrogen, etc).

If I understand correctly, I think the authors use this mixing model to distinguish differences in d15N due to depth from differences due to source (soil N vs. beaver N). 13C is used as an additional variable to allow for a third endmember. However, this

doesnt't work for several reasons. Most importantly, C and N in the same soil sample can have different sources. As a consequence of this, 13C and 15N do not necessarily show linear co-variance through the soil profile. Furthermore, it is not clear if the 15N signature of N inputs is modified as N migrates down along the soil profile. However, I don't think this mixing model is required to support the authors conclusions and I would remove is.

- Similarly, I find the $\Delta$15N values confusing and I'm not sure what they contribute to the manuscripts story. In my opinion, Fig 5a should be sufficient for report that – unlike in control soils– d15N values decrease with depth at the hotspot, representing the recent 15N-enriched N inputs from the top of the soil profile.

- It would be interesting to see a plot % beaver derived N (as in Fig 4) vs. %N (or C:N) – this would provide additional evidence that the lower C:N ratios at the hotspots have developed due to beaver N inputs.

- Would it be possible to make an estimate of the total amount of beaver-derived N retained in the soils (under a carcass) and relate that to the total amount of initial beaver N? i.e., what fraction of beaver-N is retained in the soil after 1 year?

Technical comments L47-51: this section could be more specific (e.g. use "increase/decrease"| instead of "change") L55: "insects and animals" - aren't insects animals too? L74-75: rather additional N inputsReview Keenan et al, Biogeosciences than enhanced reactions, right? L85: what's the size of the carcass (cm diameter?) - I'm wondering how much of the 60 cm diameter enrichement was located directly under the carcass L210-214: I think the main result is not a less positive slope, but rather that the linear relationship between log(%N) and d15N is lost. This makes a lot of sense as the natural processes that typically for the 15N depth gradient are masked by the recent input of 15N-enriched nitrogen. L222:"distinct isotopic enrichment" - rather distinct N sources. Enrichment is a process, not just the a differences in distinct N pools (see Z. Sharp's comments on isotope terminology

https://digitalrepository.unm.edu/unm_oer/1/ chapter 2) L297-299, 304-307: I don't really see much support for these claims for changes in biogeochemistry or discrimination in the data that is not explained by the mixing of two distinct N sources, so I would recommend removing these speculative sections. L316-318: This is a mis-interpretation of the poor linear relationship. The most shallow soil horizons have d15N value of 8.4 permil. If these horizons contain a mixture of soil and beaver N, the beaver N source signature has to be larger than 8.4 (consistent with the endmember value used in the 15N mixing model.)

---

## Referee Comment (RC5) · Michael Philben (Referee) · 2 Aug 2019

Keenan et al. use C and N stable isotope ratios to demonstrate that N derived from carrion can persist in the soil for >1 year, down to ∼10 cm depth and up to 60 cm from the site of the carcass. This shows that these decomposition hotspots can have a surprisingly long-term impact on soil nutrient status and biogeochemistry, even after visible evidence of carrion has disappeared. Previous studies have examined this question, but the present study is unique in also examining the lateral and vertical extent of carrion-derived N after 1 year.

Overall I found the paper to be interesting, concise, and easy to read. The qualitative conclusion (that carrion N can persist in the soil for >1 year) is very well supported.

[Figure]

However, I think the explanation of some of the quantitative aspects should be improved before publication.

General comments:

1. Some issues with the mixing models:

-the 2-source mixing model assumes differences in d15N are caused only by mixing of sources and are not affected by diagenetic fractionation. As noted elsewhere in the manuscript, it's quite likely that the elevated N availability would result in additional nitrification and denitrification, which would increase the d15N independent of source mixing. This assumption should be stated and its potential influence on the quantitative results discussed.

-Conversely, calculation of the isotopic discrimination factor (Figure 6) appears to ignore the impact of having a 15N-enriched source in the surface soils but not the deep soils. In other words, if the d15N depth profile is driven by distinct sources (as indicated by figures 4 and 7), then the slope in figure 6 does not represent the isotope discrimination factor

-I was confused by the use of both a 2-end member and a 3-end member mixing model. I think I understand that the former is for comparison along the lateral transect while the latter is for comparing soil profiles. Some additional explanation would be useful.

2. the introduction states a goal of ultimately moving toward quantifying ecosystem impacts of carrion inputs (Line 71). However, there is little discussion of how the results could be scaled to contribute to the ecosystem level. Can you put in context how much N was added via carrion, how much remains in the soil after 1 year, and how much was lost from the soil? It seems like this should be a relatively simple calculation using the biomass and %N of the carrion and the N content of the soils. This would be very helpful for quantifying the importance of carrion in the ecosystem N cycle.

Specific comments:

Abstract: the abstract is heavily weighted toward background information rather than results and experimental design

Lines 47-50: can you be more specific about the direction of changes observed (e.g. does pH consistently decline, etc.)?

Lines 154-157: I'm confused about the inclusion of both shallow and deep control soils in the mixing model. Can you explain the justification for this approach in more detail?

Lines 273-275: offer an explanation why carrion had not effect on d13C? (looks like the decomposition fluids had similar d13C as the surface soil)

Table 1: indicate why the 1-yr samples are bolded. N.M.=not measured?

---

## Author Response (AR1)

**Dear Dr. Pantoja,**

We appreciate the detailed and helpful comments you provided in addition to the three reviewer comments on our manuscript, "Spatial changes in soils table isotopic composition in response to carrion decomposition (BG-2018-498). We made all changes outlined in our responses to the reviewer comments, and describe below the changes in response to your comments. Please feel free to contact me if there are any questions regarding this revised submission.

Thank you,

Sarah Keenan

**AE Comments:**

1. Abstract needs major revisions (Reviewer 3's comment). Lines 1-9 are a long introduction and it does not say why this issue is relevant. Results presented here are too general to evaluate extent of this influence (for instance how big of a change in *d*15N is observed, etc.). Lines 19-21 are not very instructive: a) "...potential to result in long-term changes to soil biogeochemistry...", it is not potential, It is up to a year and of 60 cm from hot spot (already said it in lines 17-18), b) "... and to contribute to bulk soil stable isotopic composition." (already said it in lines 16-17). Instead of repeating facts, I would add significance of findings of your work for the discipline.

**Response**: We modified the abstract in response to Reviewer #3's comments, and many of the recommendations described above have been fixed. We added more specific details to the abstract describing the results of this study, and removed several lines of background text that were unnecessary and did not frame the study properly. With respect to the comment above about the text "potential to result in long-term changes", we retained this wording because currently, this is our knowledge. We know that in some soils/climates/environments, there have been clear demonstrations of long-term changes. However, this has not been demonstrated in all environments/soils, and has not been temporally resolved or spatially resolved adequately.

2. Lines 40-41. Replace microfauna in "soil microfauna (i.e., bacteria, fungi, nematodes)" since fauna refers to animals and bacteria and fungi are not.

**Response**: We retained microfauna (for nematodes) and revised the text to read: "microfauna and microbiota".

- 3. Figure 3. Label of dark circle should be sample instead of "Hotspots" **Response**: Label edited as suggested.
- 4. Paragraph of lines 236-239 repeats information from previous lines.
   Response: This text was modified in response to a reviewer's comments, and there is no longer repetition of information in the two sentences.
- 5. Line 254. Is it really 65.9%?, not 66%?

**Response**: We used the value presented by the refence cited, but we agree that 66% is perhaps more appropriate and modified the text.

6. Line 285, "to the soil profile at depth". Do you mean soil depth profile?
Response: We are referring specifically to the profile of the soil at depth. We removed "profile" and simply put "soils at depth".

7. Lines 285-287. "Decomposition hotspots, however, disrupt the expected pattern (Fig. 5), causing surface enrichment, and likely leave a lasting impact on soil stable isotopic composition." Explain what you mean with "likely leave a lasting impact on soil stable isotopic composition" since it is clear from Fig. 5 that below 10-cm depth there is no difference with respect to the control (except for one point at 30 cm depth with nitrogen stable isotopes.

**Response**: We are specifically referring to surface soils here, which are isotopically enriched and different compared to what is expected for surface soils. We are not suggesting that there are measurable changes at depth, rather that surface soils are disrupted from what is normally observed.

8. Conclusions. Please limit to conclusions of your work; Lines 323-324 and Lines 328-330 are not. Lines 331-336 are too speculative and not resulting from your data therefore do not belong to this section.

**Response**: We removed the text as suggested and include specific results/conclusions from this study.

**Dear Dr. Pantoja,**

We received three positive and constructive reviews of our manuscript, "Spatial changes in soils table isotopic composition in response to carrion decomposition (BG-2018-498)". Below we address the comments and recommendations provided by the reviewers (original comments in italics, responses beneath). We feel the revised manuscript is improved from the original version as a result of these valuable suggestions.

The primary changes to the MS include:

- Removing the three end-member model at the suggestion of Reviewer #2 (Fig. 7), which did not add to the main conclusions of the original manuscript;
- Adding details to the discussion, particularly emphasizing the broader ecological consequences of persistent carcass-enriched soil; and
- Framing the two end-member mixing model more clearly.

Other changes to the manuscript in response to specific comments are detailed below. Please feel free to contact me if there are any questions regarding this re-submission.

Thank you,

Sarah Keenan

**Anonymous Referee #1 (Received and published: 23 January 2019)**

Overall the manuscript, "Spatial changes in soil stable isotopic composition in response to carrion decomposition," within minimal revision is a well written and a sound contribution towards understanding the spatial influence of pulsed organic nutrient inputs into terrestrial ecosystems from the deposition of carrion. Numerous studies have approached this subject but the geographic expanse and complexity of the resultant biogeochemical responses leaves ample room for investigation. This work helps to bridge the gap between previous studies through both the spatial layout of the observations and the utilization of isotope discrimination factors and  $\delta^{15}N$  methodologies to tease apart the spatial extent of carrion influence within the soil profile.

Addressing the following concerns and comments will enhance the quality of this manuscript:

**Specific comments:**

1) Table 1: Insert note that defines N.M. **Response**: Note inserted to table legend.

2) Perhaps note that control values do not have an error term due to being homogenized into a single sample.

**Response**: Note inserted to table legend.

3) The table caption or a note should include the statement about this data being from Keenan et al. 2018 except for the one year data. Also include a reprint permission statement in text (section 180) and with the table if required by either journal.

**Response**: A statement has been added to clarify that some of the data is derived from Keenan et al. (2018a), and clarifying that the bolded data are new.

4) The caption states that the letters indicate differences between samples but it is not clear as to between which samples the letters are referring to from the caption or from what was readily found in the text.

**Response**: We added text to clarify that by "between samples" we were referring to between samples within each measured dataset over time (i.e., comparing pH from each sampling timepoint). The text now reads: "Letters indicate hotspot soil samples within each measured dataset (i.e., pH) that were not significantly different based on One-way ANOVA (p < 0.05)."

5) Figure 3. Again the caption states that the letters indicate differences between samples but it is not clear as to between which samples the letters are referring to from the caption or from what was readily found in the text.

**Response**: Text was added to the caption to clarify that letters indicate the soil samples taken at discrete distances from the hotspot center that were not significantly different based on a one-way ANOVA with post-hoc testing.

6) Figure 5. Similar comment to Figure 3. Clearly there are differences signified with depth but it is not readily apparent what the difference is between A, AB, B, etc.

**Response**: As with Figure 3, the caption was revised to clarify what the letters were indicating.

**Technical corrections:**

7) Section 110: The equations as written may prove confusing to readers unfamiliar with isotope ratio calculations due to the use of the backslash as the division symbol both within the numerator and denominator as well as between. Perhaps something like 13C/12Csample  $\div$  13C/12Cstandard would be better.

Response: The equation was modified as suggested.

8) Section 175: The following sentences seem to be restating a similar conclusion, "The pulse of nutrient-rich fluids resulted in significant changes to surrounding soil physio- chemistry (Table 1, Table S1). Soils exhibited long-term changes to physiochemistry following fluid degradation by soil microbial communities." Consider strengthening this paragraph by combining or differentiating these statements.

**Response**: The two sentences were combined for clarity and to eliminate redundancy.

9) Section 185: "... values around 80 cm of the hotspot" presumably should read "80cm from the hotspot".

**Response**: Text modified as suggested.

10) Section 200: Finesse this sentence a little bit to clarify that the 60 cm extent was beyond the carcass decomposition island. I believe that is what you are trying to state.

**Response**: The sentence was edited for clarity.

11) Section 235: The flow and the strength of the second sentence could be enhanced by revising the inclusion of ", here at least one year,". This is an important point that should specifically state that the results are for the given location, climate, soil, etc. and perhaps it would be better to give this its own subsequent sentence.

**Response**: The text was modified to emphasize that these results are specific for this site, a point we raise further in the discussion.

12) Typo - The third sentence, "The beaver carcasses used this study," should be "used in this study".

Response: Typo fixed.

13) Section 315: Typo - Sentence missing "a", "Based on the isotopic discrimination factor (D) for N in hotspot soils, a linear regression . . . . "

Response: Typo fixed.

Lukas Kohl (Referee #2) (Received and published: 16 July 2019) *General comments*

Keenan and co-authors investigated the effect of carrion decomposition on the underlying soil. In particular, they studied the spatial extent to a beaver carrion decomposition hotspot changed soil biogeochemical parameters (mainly C:N and d15N) one year post deposition. They find that elevated d15N values due to N inputs from the decomposing beaver were detected to 60cm lateral and 10cm depth.

The manuscript covers an important and understudied topic of terrestrial ecosystem ecology. The authors used state of the art methods and their results justify their conclusions. The manuscript reads very nicely and is surely of high interest to the Biogeosciences readership.

**Specific comments**

1) I think the main weakness of the manuscript is that the authors pooled all control samples (soils collected in some distance from the placed beavers) and analysed only a single composite sample. This means we cannot know the spatial variability of control soil properties, or the uncertainties associated with the measured average.

**Response**: We agree that pooling the control soils (a total of 5 independent locations) represents a limitation. Based on our previous studies (e.g., Cobaugh et al., 2015), we knew that the spatial and temporal variability in hotspots is far greater than that what we see in background soils. Therefore, for this experiment we collected several discrete control samples at the beginning of the experiment ("Initial" in Table 1) to assess spatial variability at the site, then a composite control sample at each time point to assess temporal variability. So, while we do not have spatial variability for each time point, we felt this combination approach was sufficient to identify the contrast between background and hotspot processes, which was the overall goal of the study.

2) The manuscript's use of biogeochemistry is somewhat confusing (e.g. L19-21). In my opinion, changes in soil d15N values may result from either changes in soil N biochemistry, or from changes in the d15N values of N inputs to soils. The manuscript's data largely suggest the latter is the dominant effect observed here. Where actual changes in the soil biogeochemistry are implied (again, e.g. L19-21), it would be better to be more specific and describe the changes in soil biogeochemistry that they think are indicated by changes.

**Response**: If we're interpreting the reviewer's comment correct, it seems they are suggesting that soil  $\delta^{15}$ N values are driven by either changes to N biogeochemistry or N inputs. However, there is scientific evidence from other systems that show that it can be combination of both – both inputs and biogeochemical process are contributing. In decomposition hotspots in particular, we know from past research that both of these processes are occurring simultaneously. The nutrient-rich carcass inputs result in enhanced microbial activity (respiration, enzyme activities, N cycling processes, etc.) and shifts in microbial communities, which have been reported in numerous studies (e.g., Macdonald et al. 2014; Cobaugh et al., 2015; Metcalf et al. 2016; Keenan et al., 2018a; Singh et al., 2018). We also directly observed elevated rates of nitrification during this decomposition study (Table 1), which suggests the N input from carcasses stimulates a microbial N cycling response. Because we measured whole system response, we cannot directly link a specific process to an enrichment effect. However, given the strong

evidence for enhanced microbial activities in this system, we have elected to retain our original explanation for the observed results: that the change in soil  $\delta^{15}$ N is driven by the carcass inputs in combination with multiple biogeochemical processes.

3) I think that assumptions that are needed for the 13C/15N three-endmember mixing model to calculate input sources for deeper soil layers are likely not met. Such a model assumes that C and N of a given soil sample originate in the same proportions from the same sources, which is not true.

**Response**: We appreciate the reviewer's thoughtful comments and agree that the assumptions of the model cannot really be met for this system. Therefore have elected to remove the three end-member mixing model from the manuscript. We initially included the model as a way to simplify the system, recognizing that in reality, as the reviewer states, this is a big assumption. Since this model was being used for simplification/illustrative purposes, removing it from the manuscript does not alter the main findings of the study.

4) Furthermore, the authors need to clarify what the mixing model actually estimates (e.g. L223: ".. evaluated the proportional contributions of three distinct sources to the stable isotopic composition in hotspot deep profiles .. ") - mixing models do not estimate contributions to the isotopic composition, but to the contribution of distinct sources to a particular pool of matter (soil organic matter, soil nitrogen, etc).

**Response**: The mixing model used (and subsequently removed in the revised MS) was originally designed to evaluate the proportional contribution of different end members (dietary sources) to a final isotopic composition (animal tissues or the "pool" of organic matter). However, we recognize the limitations of applying this trophic ecology approach towards distinguishing inputs to soil stable isotopic composition, and have removed it from the manuscript.

5) If I understand correctly, I think the authors use this mixing model to distinguish differences in d15N due to depth from differences due to source (soil N vs. beaver N). 13C is used as an additional variable to allow for a third endmember. However, this doesn't work for several reasons. Most importantly, C and N in the same soil sample can have different sources. As a consequence of this, 13C and 15N do not necessarily show linear co-variance through the soil profile. Furthermore, it is not clear if the 15N signature of N inputs is modified as N migrates down along the soil profile. However, I don't think this mixing model is required to support the authors conclusions and I would remove it.

**Response**: We completely agree with the reviewer and appreciate the suggestion to remove the three end-member mixing model from the MS. We agree that our results and conclusions are still supported by doing so.

6) Similarly, I find the  $\Delta 15N$  values confusing and I'm not sure what they contribute to the manuscripts story. In my opinion, Fig 5a should be sufficient for report that – unlike in control soils– d15N values decrease with depth at the hotspot, representing the recent 15N-enriched N inputs from the top of the soil profile.

**Response**: We included the  $\Delta^{15}$ N values as an additional way to quantify (or characterize) N changes with depth in the soil profile (lines 240-242). This approach

(subtracting soil at depth from the surface layer) calculates the 15N enrichment at each depth relative to the surface and has been used previously to identify soil profiles with perturbed N cycling or disturbed systems (e.g., Hobbie and Ouimette, 2009). These data emphasize the differences between the control and hotspot soil profiles at depth, and the consequence of local surface disturbance on calculated 15N enrichment at depth.

7) It would be interesting to see a plot % beaver derived N (as in Fig 4) vs. %N (or C:N) – this would provide additional evidence that the lower C:N ratios at the hotspots have developed due to beaver N inputs.

**Response**: Yes, we agree that this would be an interesting plot to generate, but we do not feel this plot is needed to provide additional evidence, and we do not have the data at present to accomplish this for soils at depth. Figure 3 shows that beaver-derived N (plotted as  $\delta^{15}$ N) influences soils up to 60 cm along the surface transects. The C:N values, while different within the hotspot (sample at 0 cm) compared to soil outside of the hotspot (soil at 140 cm), are not significantly different from control C:N values. There is an overall trend of lower C:N ratios within the hotspot, but because C:N does not significantly differ from control soils, we do not feel that graphing % beaver-derived N vs. C:N would add to our study.

8) Would it be possible to make an estimate of the total amount of beaver-derived N retained in the soils (under a carcass) and relate that to the total amount of initial beaver N? i.e., what fraction of beaver-N is retained in the soil after 1 year?

**Response**: Yes, this is a great suggestion. We have added this approximation to the discussion, based on the measured %N of soils relative to controls during the peak of decomposition and what was measured after one year. The text reads (Lines 279-284):

"The total %N measured in soils can be used to approximate the contribution of beaver N to soil. During active decomposition, hotspot soils contained 36 % more N compared to control soils (0.362 % N vs. 0.267 %). After one year, hotspot soils still contained 10 % more N than control soils (0.285 % N vs. 0.260 %), reflecting a loss of ~28 % of the beaver-derived N in one year."

**Technical comments:**

9) L47-51: this section could be more specific (e.g. use "increase/decrease" instead of "change")

**Response**: The text was modified as suggested. We kept reference to pH shifts in soils during decomposition to "changes" because in some soils/experiments, pH increases, while in others it decreases.

10) L55: "insects and animals" - aren't insects animals too?

Response: Yes, the reviewer is correct. We replaced "animals" with "vertebrates".

**11) L74-75: rather additional N inputs than enhanced reactions, right?**

**Response**: Decomposition hotspots exhibit changes in N due to both additional input of N (and C), which stimulates soil microbial communities and results in enhanced reaction rates.

12) L85: what's the size of the carcass (cm diameter?) - I'm wondering how much of the 60 cm diameter enrichment was located directly under the carcass

**Response**: Figure 2 provides an image of the carcass and the extent of fluid migration (the decomposition island). The soil sampled at 60 cm was not beneath the carcass (we sampled perpendicular to the carcass).

13) L210-214: I think the main result is not a less positive slope, but rather that the linear relationship between log(%N) and d15N is lost. This makes a lot of sense as the natural processes that typically for the 15N depth gradient are masked by the recent input of 15N-enriched nitrogen.

**Response**: We agree that re-phrasing our observation as a loss of the linear relationship is more appropriate and revised the text. The reviewer articulated this observation well, so we also included the explanation provided by the reviewer in the discussion.

14) L222: "distinct isotopic enrichment" - rather distinct N sources. Enrichment is a process, not just the a differences in distinct N pools (see Z. Sharp's comments on isotope terminology https://digitalrepository.unm.edu/unm oer/1/ chapter 2)

**Response**: We agree this is an important point to clarify. The text was modified as suggested, removing "distinct isotopic enrichment" and replacing it with "distinct N pools".

15) L297-299, 304-307: I don't really see much support for these claims for changes in biogeochemistry or discrimination in the data that is not explained by the mixing of two distinct N sources, so I would recommend removing these speculative sections.

**Response**: As we discuss previously in response to comment #2, there is agreement that within decomposition "hotspots" there are elevated rates of biogeochemistry, particularly N cycling. We agree that the initial input of an N source initiates changes to soil chemistry, subsequent responses by soil (and carcass-derived) microorganisms results in enhanced rates of N cycling. Given that there is support for the concept in the literature (see references cited in the response to comment #2), we do not feel that we are being overly speculative in invoking this explanation.

16) L316-318: This is a mis-interpretation of the poor linear relationship. The most shallow soil horizons have d15N value of 8.4 per mil. If these horizons contain a mixture of soil and beaver N, the beaver N source signature has to be larger than 8.4 (consistent with the endmember value used in the 15N mixing model.)

**Response**: Yes, we agree that this was a mis-interpretation (and too far-reaching) to include. We deleted the text.

**References** Cited**

- Cobaugh, K. L., Schaeffer, S. M., and DeBruyn, J. M.: Functional and structural succession of soil microbial communities below decomposing human cadavers, PLoS One, https://doi.org/10.1371/journal.pone.0130201, 2015.
- Hobbie, E. A., and Ouimette, A. P.: Controls of nitrogen isotope patterns in soil profiles, Biogeochemsitry, 95, 355-371, 2009.

- Keenan, S. W., Schaeffer, S. M., Jin, V. L., and DeBruyn, J. M.: Mortality hotspots: nitrogen cycling in forest soils during vertebrate decomposition, Soil Biol. Biochem., 121, 165-176, https://doi.org/10.1016/j.soilbio.2018.03.005, 2018a.
- Macdonald, B. C. T., M. Farrell, S. Tuomi, P. S. Barton, S. A. Cunningham, and A. D. Manning: Carrion decomposition causes large and lasting effects on soil amino acid and peptide flux. Soil Biol. Biochem., 69, 132–140, 2014.
- Metcalf, J. L., et al.: Microbial community assembly and metabolic function during mammalian corpse decomposition, Science, 351, 158–162, 2016.
- Singh et al.: Temporal and spatial impacts of human cadaver decomposition on soil bacterial and arhtropod community structure and function, Frontiers in Microbiology, https://doi.org/10.3389/fmibc.2017.02616, 2018.

**Michael Philben (Referee #3) (Received and published: 2 August 2019)**

Keenan et al. use C and N stable isotope ratios to demonstrate that N derived from carrion can persist in the soil for >1 year, down to ~10 cm depth and up to 60 cm from the site of the carcass. This shows that these decomposition hotspots can have a surprisingly long-term impact on soil nutrient status and biogeochemistry, even after visible evidence of carrion has disappeared. Previous studies have examined this question, but the present study is unique in also examining the lateral and vertical extent of carrion-derived N after 1 year.

Overall I found the paper to be interesting, concise, and easy to read. The qualitative conclusion (that carrier N can persist in the soil for >1 year) is very well supported.

However, I think the explanation of some of the quantitative aspects should be improved before publication.

**General comments:**

**1) Some issues with the mixing models:**

1A) The 2-source mixing model assumes differences in d15N are caused only by mixing of sources and are not affected by diagenetic fractionation. As noted elsewhere in the manuscript, it's quite likely that the elevated N availability would result in additional nitrification and denitrification, which would increase the d15N independent of source mixing. This assumption should be stated and its potential influence on the quantitative results discussed.

**Response**: The reviewer is completely correct, and perfectly summarized that two distinct but related processes are controlling  $\delta^{15}$ N in these soils: input of an N-rich (and enriched) source and subsequent diagenetic fractionation (driven by microbes). We have included a sentence from the reviewer's comment above into the Discussion, and added a paragraph to more clearly state that our two member mixing model likely includes contributions from both the N-enriched carcass and subsequent diagenesis.

1B) Conversely, calculation of the isotopic discrimination factor (Figure 6) appears to ignore the impact of having a 15N-enriched source in the surface soils but not the deep soils. In other words, if the d15N depth profile is driven by distinct sources (as indicated by figures 4 and 7), then the slope in figure 6 does not represent the isotope discrimination factor.

**Response**: We reworded any reference to "discrimination factor" for clarity and replaced it with "observed isotopic discrimination" to emphasize that we are not trying to make inferences about processes occurring, rather that the slope of these lines changes. As the reviewer mentions, this is driven by changes to N sources, rather than some underlying process.

1C) I was confused by the use of both a 2-end member and a 3-end member mixing model. I think I understand that the former is for comparison along the lateral transect while the latter is for comparing soil profiles. Some additional explanation would be useful.

**Response**: The reviewer is correct—the two end-member mixing model is for the surface soils and the three end-member model was used for soil profiles. However, based on comments and critiques from both Reviewer #2 and #3, we elected to remove the three end-member mixing model from the manuscript.

2) The introduction states a goal of ultimately moving toward quantifying ecosystem impacts of carrion inputs (Line 71). However, there is little discussion of how the results could be scaled to contribute to the ecosystem level. Can you put in context how much N was added via carrion, how much remains in the soil after 1 year, and how much was lost from the soil? It seems like this should be a relatively simple calculation using the biomass and %N of the carrion and the N content of the soils. This would be very helpful for quantifying the importance of carrion in the ecosystem N cycle.

**Response**: This is an excellent suggestion. Text was added to the discussion to provide these details (Lines 279-284).

**Specific comments:**

*3) Abstract: the abstract is heavily weighted toward background information rather than results and experimental design*

**Response**: The abstract was edited to remove some of the background information and to include more results.

4) Lines 47-50: can you be more specific about the direction of changes observed (e.g. does pH consistently decline, etc.)?

**Response**: This was also brought up by Referee #2 (comment 9) and the text was modified to describe the direction of changes.

5) Lines 154-157: I'm confused about the inclusion of both shallow and deep control soils in the mixing model. Can you explain the justification for this approach in more detail? **Response**: We removed the three end-member mixing model.

**6) Lines 273-275: offer an explanation why carrion had not effect on d13C? (looks like the decomposition fluids had similar d13C as the surface soil)**

**Response**: We provided a reference for Wheeler and Kavanaugh (2017), where the authors go into great detail explaining a lack of observed change in  $\delta^{13}$ C. We added a sentence to our MS to offer a brief explanation, guided by previous suggestions by Wheeler and Kavanaugh.

**7) Table 1: indicate why the 1-yr samples are bolded. N.M.=not measured?**

**Response**: Text was added to the figure legend explaining the significance of bolded data and N.M. abbreviation.

**References Cited**

[revised manuscript text omitted]

|       | Deleted: isotopic                                                                                                                                                                                                                                                                                                                                                                                                                                                                                                                                             |
|-------|---------------------------------------------------------------------------------------------------------------------------------------------------------------------------------------------------------------------------------------------------------------------------------------------------------------------------------------------------------------------------------------------------------------------------------------------------------------------------------------------------------------------------------------------------------------|
|       | Deleted: enrichment                                                                                                                                                                                                                                                                                                                                                                                                                                                                                                                                           |
| ····· | Deleted: patterns                                                                                                                                                                                                                                                                                                                                                                                                                                                                                                                                             |

[revised manuscript text omitted]
          | $\begin{array}{c} 0.299 \pm \\ 0.012^{AB} \end{array}$    | 6.79 ± 0.1 A                                | $47.83\pm5.9^{\rm AC}$                                     | 98.5 ± 0.29 A                                 | 5.11±
0.101                                    | 0.295 ±
0.009                                  | ${}^{17.33\pm}_{0.25^{AC}}$                                 | $\begin{array}{c} 1.48 \pm \\ 0.23^{\mathrm{A}} \end{array}$ | $\begin{array}{c} -27.86\pm\\ 0.08 \end{array}$   |
| Early                             | 1 August         | $\begin{array}{c} 0.216 \pm \\ 0.034^{\rm A} \end{array}$ | $\begin{array}{c} 6.86 \pm \\ 0.3^{\rm A} \end{array}$ | $73.53\pm32.2^{\rm A}$                                     | N.M.                                                     | $\begin{array}{c} 3.927 \pm \\ 1.08 \end{array}$  | $\begin{array}{c} 0.245 \pm \\ 0.058 \end{array}$ | ${}^{15.93\pm}_{1.7^{\rm AC}}$                              | $2.65 \pm 1.66^{\rm A}$                                      | $-27.73 \pm 0.42$                                 |
| Early control                     | 1 August         | 0.219                                                     | 6.82                                                   | 36.78                                                      | N.M.                                                     | 4.196                                             | 0.247                                             | 16.99                                                       | 1.76                                                         | -27.56                                            |
| Active                            | 3 August         | $\begin{array}{c} 0.234 \pm \\ 0.056^{\rm A} \end{array}$ | $\begin{array}{c} 8.64 \pm \\ 0.3^{\rm B} \end{array}$ | ${\begin{array}{c} 2150.48 \pm \\ 1282^{BC} \end{array}}$  | $\begin{array}{c} 9.16 \pm \\ 7.89^{BC} \end{array}$     | 4.251 ± 0.798                                     | $\begin{array}{c} 0.362 \pm \\ 0.096 \end{array}$ | ${\begin{array}{c} 12.01 \pm \\ 1.34^{\rm BC} \end{array}}$ | $6.23 \pm 1.50^{\rm B}$                                      | $-27.77 \pm 0.30$                                 |
| Active control                    | 3 August         | 0.160                                                     | 6.68*                                                  | 31.65*                                                     | 98.6±
1.25*                                           | 4.159                                             | 0.267                                             | 15.58*                                                      | 1.48*                                                        | -27.68                                            |
| Advanced                          | 9 August         | $\begin{array}{c} 0.286 \pm \\ 0.081^{AB} \end{array}$    | $\begin{array}{c} 8.78 \pm \\ 0.1^{\rm B} \end{array}$ | $1233\pm494^{\rm C}$                                       | $\begin{array}{c} 19.4 \pm \\ 31.2^{\rm B} \end{array}$  | $\begin{array}{c} 4.000 \pm \\ 1.29 \end{array}$  | $\begin{array}{c} 0.303 \pm \\ 0.080 \end{array}$ | $\begin{array}{c} 13.07 \pm \\ 0.75^{\rm BC} \end{array}$   | $\begin{array}{c} 8.72 \pm \\ 2.09^{\mathrm{B}} \end{array}$ | $\begin{array}{c} -27.64 \pm \\ 0.20 \end{array}$ |
| Advanced control                  | 9 August         | 0.223                                                     | 6.84*                                                  | 43.42*                                                     | $\begin{array}{c} 98.0 \pm \\ 0.57 \ast \end{array}$     | 5.008                                             | 0.281                                             | 17.82*                                                      | 1.26*                                                        | -27.77                                            |
| Early
skeletal                 | 6 September      | $\begin{array}{c} 0.242 \pm \\ 0.070^{\rm A} \end{array}$ | $\begin{array}{c} 7.58 \pm \\ 0.4^{\rm C} \end{array}$ | ${\begin{array}{c} 973.8 \pm \\ 211^{\rm AC} \end{array}}$ | $\begin{array}{c} 97.4 \pm \\ 0.84^{\rm AC} \end{array}$ | $\begin{array}{c} 3.610 \pm \\ 0.839 \end{array}$ | $0.293 \pm 0.057$                                 | ${}^{12.28\pm}_{0.74^{BC}}$                                 | $\begin{array}{c} 9.26 \pm \\ 1.54^{\rm B} \end{array}$      | $\begin{array}{c} -27.74 \pm \\ 0.30 \end{array}$ |
| Early
skeletal
control      | 6 September      | 0.121                                                     | 6.84*                                                  | 35.08                                                      | $98.3\pm0.50$                                            | 4.023                                             | 0.259                                             | 15.53*                                                      | 1.78*                                                        | -27.63                                            |
| Late skeletal                     | 9 December       | $\begin{array}{c} 0.271 \pm \\ 0.021^{AB} \end{array}$    | $\begin{array}{c} 6.93 \pm \\ 0.3^{\rm A} \end{array}$ | ${}^{225.2\pm}_{84.8^{\rm AC}}$                            | $100\pm0^{\rm A}$                                        | $2.668 \pm 0.352$                                 | $\begin{array}{c} 0.214 \pm \\ 0.030 \end{array}$ | ${\begin{array}{c} 12.51 \pm \\ 0.67^{BC} \end{array}}$     | $9.25 \pm 1.33^{\mathrm{B}}$                                 | -27.41 ± 0.25                                     |
| Late skeletal
con t rol | 9 December       | 0.246                                                     | 6.73                                                   | 29.13                                                      | $100\pm0$                                                | 2.084                                             | 0.136                                             | 15.37*                                                      | 1.79*                                                        | -27.30                                            |
| 1 yr. post
decay               | 10 August        | $0.404 \pm 0.027^{\rm B}$                                 | 6.10 ± 0.3 D                                | $\textbf{29.47} \pm \textbf{7.6}^{A}$                      | N.M.                                                     | 4.253 ± 1.07                                      | 0.285 ±
0.036                                  | 15.24 ± 3.49 °                                   | $\begin{array}{c} 8.42 \pm \\ 1.52^{\mathrm{B}} \end{array}$ | -27.67 ± 0.25                                     |

yr. post decay, were previously published in Keenan et al. (2018a).

26

De

De

For

| 1 yr. post
decay | 10 August | 0.449* | 6.29* | 23.07 | N.M. | 4.36 | 0.26 | 17.08 | 0.05* | -27.73 |
|---------------------|-----------|--------|-------|-------|------|------|------|-------|-------|--------|
| control             |           |        |       |       |      |      |      |       |       |        |

**Deleted: ¶**

**Table 1 (continued):**

I

|                                   |                                                   | Microbial
respiration                                                         |                                                                             |                                                               | Nitrification
potential                                           |                                                           |                                                               |                                     |
|-----------------------------------|---------------------------------------------------|----------------------------------------------------------------------------------|-----------------------------------------------------------------------------|---------------------------------------------------------------|----------------------------------------------------------------------|-----------------------------------------------------------|---------------------------------------------------------------|-------------------------------------|
|                                   | Protein (mg
g -1 )                  | rate (µg CO 2 -
C release
gdw -1 day -1 ) | DOC (μg
C gdw -1 )                                            | Ammonium
(mg NH4-N
gdw -1 )                  | rate (mg
NO 2 gdw -1
day -1 ) | Nitrate (mg
NO3 - -N
gdw -1 ) | DON (mg N
gdw -1 )                              | Accumulated
Degree Days
(ADD) |
| Initial                           | $\begin{array}{c} 0.251 \pm \\ 0.051 \end{array}$ | $50.9\pm8^{\rm A}$                                                               | $\begin{array}{c} 2.44 \pm \\ 0.2^{\rm AC} \end{array}$                     | $\begin{array}{c} 0.039 \pm \\ 0.01^{\rm A} \end{array}$      | ${\begin{array}{c} 0.181 \pm \\ 0.05^{\rm A} \end{array}}$           | $0.000\pm0.0^{\rm A}$                                     | ${\begin{array}{c} 0.283 \pm \\ 0.02^{\rm A} \end{array}}$    | 26.1                                |
| Early                             | $\begin{array}{c} 0.200 \pm \\ 0.037 \end{array}$ | $51.4\pm18^{\rm A}$                                                              | $\begin{array}{c} 3.44 \pm \\ 1.0^{A} \end{array}$                          | $\begin{array}{c} 0.101 \pm \\ 0.06^{\mathrm{A}} \end{array}$ | $\begin{array}{c} 0.237 \pm \\ 0.16^{\mathrm{A}} \end{array}$        | $0.000\pm0.0^{\rm A}$                                     | $\begin{array}{c} 0.718 \pm \\ 0.35^{\text{A}} \end{array}$   | 106.7                               |
| Early control                     | $\begin{array}{c} 0.159 \pm \\ 0.004 \end{array}$ | 35.2                                                                             | 2.35                                                                        | 0.011                                                         | 0.167                                                                | 0.000                                                     | 0.242                                                         | 106.7                               |
| Active                            | $\begin{array}{c} 0.260 \pm \\ 0.025 \end{array}$ | $300\pm90^{\rm B}$                                                               | $\begin{array}{c} 66.54 \pm \\ 40.4^{\mathrm{B}} \end{array}$               | $2.49\pm1.24^{\rm B}$                                         | $\begin{array}{c} 0.366 \pm \\ 0.09^{\mathrm{A}} \end{array}$        | $0.001\pm0.0^{\rm A}$                                     | $1.363\pm1.4^{\rm A}$                                         | 160.3                               |
| Active control                    | $\begin{array}{c} 0.225 \pm \\ 0.007 \end{array}$ | 27.8*                                                                            | 2.52*                                                                       | 0.010*                                                        | 0.163                                                                | 0.000                                                     | 0.205                                                         | 160.3                               |
| Advanced                          | $0.281 \pm 0.072$                                 | $162.8\pm110^{\rm A}$                                                            | $42.5 \pm 30.0^{\circ}$                                                     | $\begin{array}{c} 2.29 \pm \\ 1.80^{BC} \end{array}$          | $\begin{array}{c} 0.517 \pm \\ 0.17^{\mathrm{A}} \end{array}$        | $0.001\pm0.0^{\rm A}$                                     | $1.906 \pm 1.7^{\rm A}$                                       | 321.7                               |
| Advanced control                  | $0.239 \pm 0.024*$                                | 45.7*                                                                            | 3.44*                                                                       | 0.015*                                                        | 0.181                                                                | 0.003                                                     | 0.304                                                         | 321.7                               |
| Early
skeletal                 | $\begin{array}{c} 0.238 \pm \\ 0.049 \end{array}$ | $76.9\pm42^{\rm A}$                                                              | $14.3 \pm 3.4^{\rm AC}$                                                     | $\begin{array}{c} 0.775 \pm \\ 0.14^{\rm AC} \end{array}$     | $8.57\pm4.4^{\rm B}$                                                 | $\begin{array}{c} 0.309 \pm \\ 0.169^{\rm B} \end{array}$ | $\begin{array}{c} 6.089 \pm \\ 1.24^{\rm B} \end{array}$      | 1042.8                              |
| Early
skeletal
control      | $\begin{array}{c} 0.193 \pm \\ 0.021 \end{array}$ | 20.8                                                                             | 2.89                                                                        | 0.006                                                         | 0.130*                                                               | 0.000*                                                    | 0.185*                                                        | 1042.8                              |
| Late skeletal                     | $\begin{array}{c} 0.250 \pm \\ 0.014 \end{array}$ | $57.2\pm26^{\rm A}$                                                              | $\begin{array}{c} 10.2 \pm \\ 8.4^{\rm AC} \end{array}$                     | $\begin{array}{c} 0.246 \pm \\ 0.06^{\rm A} \end{array}$      | $\begin{array}{c} 0.017 \pm \\ 0.20^{\mathrm{A}} \end{array}$        | $\begin{array}{c} 0.019 \pm \\ 0.001^{\rm A} \end{array}$ | $\begin{array}{c} 1.929 \pm \\ 0.37^{\mathrm{A}} \end{array}$ | 2591.7                              |
| Late skeletal
con t rol | $\begin{array}{c} 0.195 \pm \\ 0.016 \end{array}$ | 46.5                                                                             | 3.12                                                                        | 0.012                                                         | 0.039                                                                | 0.000                                                     | 0.309                                                         | 2591.7                              |
| 1 yr. post
decay               | 0.239 ± 0.021                                     | $64.1\pm8^{\rm A}$                                                               | $\begin{array}{c} \textbf{2.81} \pm \\ \textbf{0.5}^{\text{A}} \end{array}$ | $\boldsymbol{0.008 \pm 0.0^{\mathrm{A}}}$                     | $0.006 \pm 0.00^{\mathrm{A}}$                                        | $0.001\pm0.0^{\rm A}$                                     | $0.095 \pm 0.01^{\rm A}$                                      | 6377.5                              |
| 1 yr. post
decay
control    | 0.195±0.008                                       | 59.5                                                                             | $\textbf{2.43} \pm \textbf{0.0}$                                            | 0.007                                                         | 0.006                                                                | 0.000                                                     | 0.093                                                         | 6377.5                              |

27

Table 2: Differences in soil  $\delta^{15}N$  at depth and  $\delta^{15}N$  in surface soils for hotspot and control depth

**profiles.**

|               | $\Delta^{15}$ N (‰) |         |
|---------------|---------------------|---------|
| Depth
(cm) | Hotspot             | Control |
| 0             | 0                   | 0       |
| 5             | -2.1                | 2.65    |
| 10            | -1.5                | 3.2     |
| 15            | 0.1                 | 6.6     |
| 20            | 0.9                 | 7.7     |
| 30            | 0.9                 | 6.1     |
| 40            | 1.2                 | 8.4     |

660

---

## Editor Decision (ED1)

Review of bg-2018-498

August 30, 2019
Dear Dr. Keenan and Dr. DeBruyn,
Thanks for providing responses to reviewers of ms bg-2018-498. I am confident that those responses will be included in the new version. In addition to that, I would like you to considering the following issues:

1. **Abstract** needs major revisions (Reviewer 3´s comment). Lines 1-9 are a long introduction and it does not say why this issue is relevant. Results presented here are too general to evaluate extent of this influence (for instance how big of a change in $\delta^{15}N$ is observed, etc.). Lines 19-21 are not very instructive: a) "…potential to result in long-term changes to soil biogeochemistry…", it is not potential, It is up to a year and of 60 cm from hot spot (already said it in lines 17-18), b) "… and to contribute to bulk soil stable isotopic composition." (already said it in lines 16-17). Instead of repeating facts, I would add significance of findings of your work for the discipline.

2. Lines 40-41. Replace microfauna in "soil microfauna (i.e., bacteria, fungi, nematodes)" since fauna refers to animals and bacteria and fungi are not.

3. Figure 3. Label of dark circle should be sample instead of "Hotspots"
4. Paragraph of lines 236-239 repeats information from previous lines.
5. Line 254. Is it really 65.9%?, not 66%?
6. Line 285, "to the soil profile at depth ". Do you mean soil depth profile?
7. Lines 285-287. "Decomposition hotspots, however, disrupt the expected pattern (Fig. 5), causing surface enrichment, and likely leave a lasting impact on soil stable isotopic composition." Explain what you mean with "likely leave a lasting impact on soil stable isotopic composition" since it is clear from Fig. 5 that below 10-cm depth there is no difference with respect to the control (except for one point at 30 cm depth with nitrogen stable isotopes.

8. Conclusions. Please limit to conclusions of your work; Lines 323-324 and Lines 328-330 are not. Lines 331-336 are too speculative and not resulting from your data therefore do not belong to this section.

I am looking forward to receive a new version of your article

Sincerely yours

Silvio Pantoja
Associate Editor